# Adjusting Client-Level Risks Impacts on Home Care Organization Ranking

**DOI:** 10.3390/ijerph18115502

**Published:** 2021-05-21

**Authors:** Aylin Wagner, René Schaffert, Julia Dratva

**Affiliations:** 1Institute of Health Sciences, ZHAW Zurich University of Applied Sciences, Katharina-Sulzer-Platz 9, 8401 Winterthur, Switzerland; rene.schaffert@zhaw.ch (R.S.); julia.dratva@zhaw.ch (J.D.); 2Department of Health Sciences and Medicine, University of Lucerne, Frohburgstrasse 3, 6002 Lucerne, Switzerland; 3Medical Faculty, University of Basel, Klingelbergstrasse 61, 4056 Basel, Switzerland

**Keywords:** risk adjustment, quality indicators, home care, quality of health care, urinary incontinence

## Abstract

Quality indicators (QIs) based on the Resident Assessment Instrument-Home Care (RAI-HC) offer the opportunity to assess home care quality and compare home care organizations’ (HCOs) performance. For fair comparisons, providers’ QI rates must be risk-adjusted to control for different case-mix. The study’s objectives were to develop a risk adjustment model for worsening or onset of urinary incontinence (UI), measured with the RAI-HC QI bladder incontinence, using the database HomeCareData and to assess the impact of risk adjustment on quality rankings of HCOs. Risk factors of UI were identified in the scientific literature, and multivariable logistic regression was used to develop the risk adjustment model. The observed and risk-adjusted QI rates were calculated on organization level, uncertainty addressed by nonparametric bootstrapping. The differences between observed and risk-adjusted QI rates were graphically assessed with a Bland-Altman plot and the impact of risk adjustment examined by HCOs tertile ranking changes. 12,652 clients from 76 Swiss HCOs aged 18 years and older receiving home care between 1 January 2017, and 31 December 2018, were included. Eight risk factors were significantly associated with worsening or onset of UI: older age, female sex, obesity, impairment in cognition, impairment in hygiene, impairment in bathing, unsteady gait, and hospitalization. The adjustment model showed fair discrimination power and had a considerable effect on tertile ranking: 14 (20%) of 70 HCOs shifted to another tertile after risk adjustment. The study showed the importance of risk adjustment for fair comparisons of the quality of UI care between HCOs in Switzerland.

## 1. Introduction

In Switzerland, as in many other countries, healthcare is under increased scrutiny to enhance the quality and safety of patients. While quality measures for hospital-based care are well established and publicly reported, reliable and accessible information on the quality of home care is lacking [1]. This gap is ever more pronounced, because home care is becoming increasingly important due to an ageing population and a rise in prevalence of chronic conditions.

The Swiss healthcare system is highly decentralized with 26 cantons (federal states) and 2352 municipalities [2]. Home care is organized according to the canton and municipal policy and services are provided by nonprofit and private home care organizations (HCOs) as well as self-employed nursing professionals. In 2019, almost 395,000 persons (4.6% of the total population) received home care, with 80% of clients receiving services from nonprofit HCOs [3]. HCOs in Switzerland vary in size and range from small rural to large urban organizations. Nonprofit HCOs tend to be larger than private HCOs partly due to concentration processes to increase efficiency [4]. Home care aims at maintaining and stimulating the autonomy of the client and comprises nursing services as well as household tasks. Depending on the region and organization, other services such as home care for children, palliative care, and meal service are provided [5]. Responsibility for home care services lies at the level of the cantons or municipalities. Accordingly, local governments formulate a performance mandate for nonprofit HCOs to ensure help and care at home. HCOs with a performance mandate have the obligation to provide services to all clients and therefore cannot refuse clients [4].

National data on home care quality does not exist in Switzerland, although there is a legal basis that obliges providers to report data on clinical quality indicators (QIs) to the respective federal authorities [6]. The law is not being implemented due to a lack of knowledge which home care QIs are appropriate to measure quality of home care. To date, the only existing home care QIs are outcome indicators based on the Resident Assessment Instrument-Home Care (RAI-HC) Schweiz, an instrument widely used in Switzerland and internationally for home care planning. The validity and reliability of the RAI-HC have been tested in several studies showing overall good results [7,8,9,10]. The Swiss RAI-HC QIs have so far only been used for internal quality improvement processes. However, in a recent QI evaluation study [11,12], the indicators were examined for their appropriateness for national quality measurement.

Since health outcomes are not only a function of quality of care but also depend on the health status of patients [13], fair national comparisons of home care quality require risk adjustment that statistically accounts for differences in the mix of clients’ risk across HCOs [14]. Therefore, risk adjustment makes comparisons of outcomes across providers more meaningful [14]. Two main approaches to risk adjustment are found in the literature: stratification and indirect standardization [15,16,17]. Older risk adjustment approaches for QIs used stratification, i.e., stratifying the population by a risk factor (e.g., age, sex) and calculating QIs separately within each group, considering only one risk factor at a time [17]. A strength of stratification is computational simplicity; however, the approach is problematic when providers do not have enough cases to produce reliable and stable estimates of QI rates within each risk group [15]. Contemporary risk adjustment often uses indirect standardization applying regression models to derive estimates of an individual client’s likelihood of experiencing an outcome [17]. Based on the predicted values, expected QI rates are generated and compared to observed QI rates, e.g., by calculating the observed-to-expected ratio (O/E ratio) [16,18]. An advantage of indirect standardization is that multiple risk factors are taken into account simultaneously in the adjustment process, in contrast to stratification [17]. The identification of risk factors is relevant in each approach, including the description of the clinical relationship of the factor with the measured outcome and whether the factors can be influenced by providers [16,19]. However, it is important not to adjust for factors that reflect poor quality of care [16,20].

While risk adjustment for international RAI-HC QIs has been investigated in a few studies [21,22,23], risk adjustment for the Swiss RAI-HC QI has rarely been examined. The only study conducted in 2015 using indirect standardization was limited by small sample size, resulting in risk adjustment models with poor performance [24]. The risk adjustment approach developed for RAI-HC QIs is based on stratification and regression modelling within each strata [23]. This approach is not applicable to the Swiss home care setting as Switzerland uses an adapted and shorter version of the RAI-HC and comparatively Swiss HCOs are smaller resulting in smaller sample sizes.

Since 2016, the database HomeCareData (HCD) [25,26] has been in existence for calculating Swiss RAI-HC QIs. HCD centralizes client-level data collected with the RAI-HC and is currently the only register in Switzerland that contains client-level home care data. The database is still in the development phase, but the number of data suppliers is constantly growing. In 2020, 104 of about 400 nonprofit HCOs transmitted data to HCD [25].

In this study, we explore the potential and the limitations of HCD for risk adjustment of the Swiss RAI-HC QIs by taking the outcome “worsening or onset of urinary incontinence (UI)”, measured with the RAI-HC QI bladder incontinence, as an exemplary QI. We chose the QI bladder incontinence because UI implies a substantial economic burden for the health care system and is a major health problem for home care clients with a profound influence on physical and psychosocial well-being [27,28]. Furthermore, the QI addresses not only a relevant outcome but also a common quality problem. Worsening or onset of UI affects 11% of Swiss home care clients, and QI rates vary considerably between HCOs [29]. Nevertheless, UI is preventable and treatable. Evidence-based guidelines for the management of UI in older adults indicate that behavioral interventions (e.g., bladder training, pelvic floor exercises) and pharmacotherapy can reduce UI [30,31]. Therefore, home care nurses can play a pivotal role in the quality of UI care [32]. The manuscript presents results on the development of the risk adjustment model for the QI bladder incontinence using HCD data and the impact of risk adjustment on quality rankings of HCOs.

## 2. Materials and Methods

### 2.1. Study Data

The study uses the HCD [25,26], a database centralizing client-level data collected with the RAI-HC in Switzerland. The transmission of data by nonprofit HCOs to HCD is not mandatory; therefore, the register only contains data from 76 of 392 nonprofit HCOs (as of 2018). RAI-HC assessments are conducted by home care nurses at client intake and then every three to six months or when there is a significant change in clients’ health status. Thus, it is possible to follow clients over time and assess changes in health status, such as the worsening or onset of UI. HCD contains some socio-demographic data, clinical data as well as information on care needs and planned services. For this study, we extracted and cleaned HCD data from 1 January 2017 to 31 December 2018. We excluded home care clients younger than 18 years old and clients who had only one RAI-HC assessment in the selected time span because two assessments are needed to measure change in UI. In addition, we excluded clients who were terminally ill since QI bladder incontinence was not developed for clients with palliative care needs.

### 2.2. Construction of the Quality Indicator

The outcome for our model was the RAI-HC QI bladder incontinence, i.e., worsening or onset of UI. Worsening or onset of UI was measured with the variable “Bladder incontinence–ability to control the bladder in the last 3 days” (continent/continent with catheter or stoma/mostly incontinent/incontinent with residual control/incontinent). Based on the UI status in two assessments (pre-assessment and follow-up assessment), we constructed a dichotomous variable and categorized clients in two groups: Group one comprises clients that experienced worsening or onset of UI between the assessments (variable set to 1) and clients experienced no change or improvement in UI (variable set to 0).

The QI is described with a numerator (number of clients with the outcome) and denominator (number of clients at risk for the outcome and not otherwise excluded from the QI), i.e., it expresses a rate of home care clients that experienced worsening or onset of UI. The higher the QI rate, the more clients experienced worsening or onset of UI. Table 1 describes the QI. The QI is measured over one calendar year, i.e., for each client, we selected the most recent follow-up assessment in 2018 and its pre-assessment. The interval between the two assessments had to be no less than 30 days and no more than 365 days, otherwise, the case was excluded.

### 2.3. Identification and Construction of Risk Factors

We identified potential client-level risk factors associated with worsening or onset of UI by conducting a non-systematic literature review in the electronic databases PubMed and CINAHL. An overview of the identified risk factors is reported in Appendix A. Risk factors were considered for the modelling of risk adjustment if they were available in the HCD and not influenceable by the provider. The risk factors were constructed from data reported in the pre-assessment reflecting the clients’ health status before the potential worsening or onset of UI.

The following risk factors were included in the model: age (18–64, 65–79, ≥80), sex (female/male), body mass index (BMI, <25 kg/m^2^, 25–29.9 kg/m^2^, ≥30 kg/m^2^), daily smoking (yes/no), Cognitive Performance Scale (CPS, scale of 0–3) [33], Activities of Daily Living (ADL) index (includes “bed movement”, “transferring”, “toilet use”, and “eating”) [34], impairment in locomotion (yes/no), impairment in personal hygiene (yes/no), impairment in bathing (yes/no), unsteady gait (yes/no), Instrumental Activities of Daily Living (IADL) scale (includes “meal preparation”, “general housework”, “manage money”, “handling medication”, “making a call”, and “shopping”; scale of 0–6) [34], Depression Rating Scale (DRS, scale of 0–3) [35], bowel incontinence (yes/no), hospitalization in the last 90 days (yes/no) and type of pre-assessment (intake/significant change in health status/regular follow-up) as proxy measures of deterioration of health status, and time between assessments (30–180 days/181–365 days).

### 2.4. Statistical Analysis

In this study, we applied the risk adjustment approach of indirect standardization [16,18], which involved the following steps: first, the development of a risk adjustment model that allows the prediction of QI rates. Second, the generation of observed (O) and (model-predicted) expected QI rates (E) to calculate the O/E ratio, which was then multiplied by the total QI rate (client population mean) to obtain risk-adjusted QI rates. The procedure is described in more detail below.

Logistic regression was used to develop a risk adjustment model for the outcome worsening or onset of UI. We randomly split the dataset into a development dataset (80% of cases, *n* = 10,122 clients) and validation dataset (20% of cases, *n* = 2530 clients). The models (and the QI rates) were constructed using the development dataset and the accuracy of the selected model was assessed using the validation data set. We performed exploratory univariate analyses of the associations between each potential risk factor and worsening or onset of UI, followed by multivariable logistic regression analysis predicting the probability of worsening or onset of UI for each home care client, conditional on his/her risk at the pre-assessment. Variables with a *p* value ≤ 0.25 in univariate analyses were included in the multivariable model. First, we performed a full model including all risk factors, followed by a reduced model, in which risk factors that were non-significant (*p* > 0.1) were excluded. Model performance of both models was tested using the concordance statistic (c-statistic) for discriminative ability [36]. The Akaike Information Criterion (AIC) [37] was calculated to compare the full and reduced model and to select the best-fit model, which was the reduced model. This model we applied to the validation data to assess the discriminatory performance of the model in the validation data set via the c-statistic.

After fitting the logistic regression model, we calculated the risk-adjusted QI rates based on the development data set for the HCOs. We restricted the calculations to HCOs with ten or more cases in the denominator to obtain relatively stable estimates. For each HCO, we computed the actual observed QI rate (O = number of clients with worsening or onset of UI/number of clients treated) and the expected QI rate, adjusted for the case-mix (E = total sum of predicted worsening or onset of UI/number of clients treated). The risk-adjusted rate was determined as the ratio of observed-to-expected rate (O/E ratio) multiplied by the total QI rate (client population mean) [18,22]. Such risk-adjusted QI rates show which QI rates would have been obtained if all HCOs would have the same client mix. Thus, the QI rates are directly comparable [38].

To estimate uncertainty in the risk-adjusted QI rates, we used nonparametric bootstrapping considering the nesting of the clients in the HCOs. We created 10,000 samples and gained an estimate of the empirical distribution of the risk-adjusted QI rates on organization level. The distribution limits were calculated at the 2.5% and 97.5% percentiles to generate 95% confidence intervals (CIs) for the risk-adjusted QI rates for each HCO. The 95% CIs were compared to the total QI rate (client population mean) to assess which HCOs were statistically different.

We explored the difference between observed and risk-adjusted QI rates graphically using a Bland-Altman plot [39]. In the Bland-Altman plot, the difference between the observed and risk-adjusted QI rates is plotted against the average of the two measurements. Furthermore, we ranked the HCOs by observed and risk-adjusted QI rates and then split observed and adjusted QI rates into tertiles. We assessed the change in tertile rank after risk adjustment, i.e., we examined the number of HCOs that shifted to a better or worse tertile. We chose tertiles because the ranking in three groups assigned to three performing levels (high, middle, low) is easy to interpret and thus meaningful to HCOs and clients [40,41]. All statistical analyses were completed using Stata Version 15 (StataCorp LLC, College Station, TX, USA).

## 3. Results

### 3.1. Sample Characteristics

In total, 12,652 home care clients from 76 HCOs were included in the study, i.e., they had at least two RAI-HC assessments of which the follow-up was conducted in 2018. 87.7% of the clients were aged 65 years and older with a mean age (SD) in the overall sample of 79.5 (12.9) years and 63.9% of the clients were female (Table 2). 28.6% of the clients were urinary incontinent (women 32.0%, men 22.5%) in the pre-assessment. 10.7% of the clients experienced worsening or onset of UI from the pre-assessment to follow-up assessment.

QI rates on organization level were calculated for 70 of the 76 HCOs because six HCOs had less than ten cases in the denominator, i.e., provided services to less than ten clients, and were excluded. The 70 HCOs are from 20 (of 26) cantons, represent three language regions of Switzerland (German, French, and Italian speaking parts), and provided care to an average of 144 clients (SD 247, Min. 10, Max. 1915). Characteristics of the HCOs are reported in Appendix B.

### 3.2. Risk Adjustment Model

The multivariable logistic regression yielded a model with eight significant (*p* ≤ 0.05) client-level predictors of worsening or onset of UI (Table 3): older age, female sex, obesity, impairment in cognition, impairment in hygiene, impairment in bathing, unsteady gait, and hospitalization. In the model, “type of pre-assessment” showed a non-significant association. The model achieved a c-statistic of 0.658 (CI 0.638–0.673). The c-statistic in the validation data set was not significantly different with a value of 0.685 (CI 0.653–0.716).

### 3.3. Risk-Adjusted Quality Indicator Rates

Figure 1 displays the 70 HCOs ranked by their risk-adjusted QI rates and 95% CIs from the bootstrap resampling distribution. The 95% CIs are relatively large and influenced by the denominator size, i.e., the case number (clients) of the HCO. Large HCOs with relatively high case numbers have narrower CIs than small HCOs with low case numbers. The graph indicates that 12 (17,1%) of the 70 HCOs are statistically different from the average, as their CIs do not intersect the client population mean (total QI rate). HCOs with 95% CIs below the total QI rate (blue line) have better than average quality performance, and vice versa.

### 3.4. Comparison of Observed and Risk-Adjusted Quality Indicator Rates

The observed QI rates ranged from 0% to 26.92% (mean = 11.70%, SD = 5.44, IQR = 7.69–14.62) and the risk-adjusted QI rates ranged from 0% to 26.67% (mean = 11.65%, SD = 5.04, IQR = 8.28–13.92).

The Bland-Altman plot was used to explore graphically the differences between the observed and risk-adjusted QI rates (Figure 2). The mean difference between the observed and risk-adjusted QI rates is 0.05% with 95% limits of agreement (mean ± 2 SD) of −3.03% to 3.13%. The plot shows that six (8.57%) of 70 HCOs fall outside the 95% limits of agreement (four HCO above and two HCO below the lower 95% limit) and are outliers.

The HCOs were ranked based on the observed and risk-adjusted QI rates and then grouped into tertiles. 14 (20%) of 70 HCOs shifted into another tertile after risk adjustment, of which seven HCOs shifted to a lower tertile (i.e., better performance) and seven HCOs to a higher tertile (i.e., worse performance). Four (16.7%) of 24 HCOs in the lower tertile shifted into the middle tertile. Three (13.0%) of 23 HCOs in the middle tertile shifted into the upper tertile after risk adjustment, and four (17.4%) shifted into the lower tertile. Three (13.0%) of 23 HCOs in the upper tertile shifted into the middle tertile after risk adjustment.

## 4. Discussion

In this study, we developed a risk adjustment model based on HCD data taking RAI-HC QI bladder incontinence as an exemplary QI. The model yielded eight risk factors significantly associated with worsening or onset of UI and met statistical standards of goodness-of-fit. Risk adjustment demonstrated substantial impact on quality rankings of HCOs. One of five HCOs shifted to another tertile after risk adjustment, i.e., without adjusting for case-mix, the tertile ranking of 20% of 70 HCOs would have been incorrect, either too high or too low.

These findings indicate the importance for risk-adjusting the QI bladder incontinence for fair comparisons of quality of UI care between HCOs. It must be noted that different ranking methods can yield different results. The impact of risk adjustment on HCOs ranking would have been larger if we have chosen to divide the organizations into more intervals than tertiles, as less change would have been needed to shift to another group. Deutscher et al. [41], for example, analyzed the impact of risk adjustment on rankings of outpatient clinics and categorized the clinics in deciles as well as three distinct quality groups. They found that 70% of clinics changed decile rank and 31% shifted to another quality group following risk adjustment [41].

Furthermore, it must be pointed out that the HCO ranking does not inform on actual quality of home care with respect to UI. The ranking presents the relative QI performance among the HCOs and with respect to the client population mean (total QI rate). An established benchmark of best practice or standards for quality of home care for the QI bladder incontinence are currently lacking in Switzerland and also internationally [1,42]. Thus, it remains unclear, even after risk-adjustment, which QI rate corresponds to high quality care or potential care problems. For nursing home QIs, Hjaltadóttir et al. [43] developed thresholds indicating areas of good and poor care in Iceland. Together with an expert panel, she defined for the QI “bladder or bowel incontinence” a lower threshold and an upper threshold, 35.4% and 64.3% respectively. Such thresholds (or reference ranges within the benchmark) must also be defined for the Swiss RAI-HC QIs to provide organizations with quality aims and indications of the quality of care they provide. According to Rantz et al. [44], thresholds are also relevant on a public health level to reinforce excellent performance (e.g., with pay-for-performance programs) and flag potential problem areas.

In our study, the impact of risk adjustment on organization level was substantial. To our knowledge, no other study has investigated the impact of risk adjustment of the RAI-HC QI bladder incontinence on HCO rankings with the similar rigor. Schaffert and Staub [24], for example, developed risk adjustment models for several Swiss RAI-HC QIs in 2015 without addressing the QI specific risk factors, resulting in poor performance of the models. Internationally, research has been conducted on regional and national level comparing observed and risk-adjusted RAI-HC QIs [21,22,23,45,46,47]. Overall, and similar to our results on organization level, the findings showed moderate effects of risk adjustment, mostly slightly reducing differences in QI rates between regions or countries. In addition, we cannot compare the performance of our risk adjustment model because the studies that looked at developing risk adjustment models for RAI-HC QIs [21,22,23] did not publish the logistic regression models or information on the statistical performance of the models.

### 4.1. Strength and Limitations

This study contributes to the sparse literature on risk adjustment methods for RAI-HC QIs by developing a risk adjustment model for the RAI-HC QIs bladder incontinence and highlighting the impact of risk adjustment on rankings. One of the main strengths of this study is that prior to statistical modelling, we identified client-level risk factors for worsening or onset of UI in the scientific literature and illustrated the clinical relationship of the factors with UI. In addition, to prevent over-adjustment, i.e., unwittingly adjust away poor care practice, we carefully considered for each potential risk factor whether it was truly a measure of risk or a condition that home care nurses could influence by providing appropriate care [22]. For example, we chose to exclude the risk factors’ delirium and obstipation because evidence-based guidelines indicate that nurses can adequately manage these health problems [48,49] and that these factors therefore rather reflect poor care quality than risks. It is, however, debatable which factors can be influenced by nurses and to what extent, especially in home care, characterized by less control over outcomes than institutional care in hospitals or nursing homes [50].

A general limitation of this study was data availability. While we purposefully excluded some risk factors, other factors could not be operationalizable with the HCD data due to missing items (e.g., medication) or open text fields (e.g., diagnoses, environmental barriers) in the Swiss RAI-HC. This is a limitation of our model, because diagnoses and comorbidity are commonly included in risk adjustment models [16,19] and scientific evidence shows that neurological diseases and diabetes mellitus significantly increase the risk of developing UI [51,52,53]. We believe that the inclusion of such predictors in the risk adjustment model would most likely result in a model with an even better performance.

Further, a selection bias may have occurred since our sample consisted only of data from 76 (19.4%) of 392 nonprofit HCOs in 2018, which limits the generalization of the results and implies cautious interpretation.

### 4.2. Implication for Research and Policy

While this study has shown that the database HCD yields the potential for thorough risk adjustment of the RAI-HC QIs, lack of routine data of high quality are crucial limitations [1]. Due to the implementation of the interRAI-HC in Switzerland (longer version of the RAI-HC) in 2020, an increase in HCD data suppliers and improved data quality are expected. Future research on the risk adjustment of the QI bladder incontinence should examine further development of the risk adjustment model by including additional risk factors collected with the new interRAI-HC. Furthermore, a comprehensive national effort is needed to improve data on the quality in home care and to establish a national register comprising representative home care data [1].

On a policy level, efforts are needed to define a home care QI set for publicly reporting. The statutory mandate to measure quality of health care lies with the Federal Office of Public Health (FOPH), as well as the decision on which home care QIs shall be reported. In this paper, we only investigated one RAI-HC QIs. However, quality is a multidimensional concept and home care quality should be reflected with a large number of indicators [42,54]. Although it requires considerable effort as well as clinical and methodological knowhow [55], we highly recommend to publicly report a set of different risk-adjusted QIs and to develop evidence-based risk adjustment models following the steps presented in this paper. Finally, a solution should be found on how to include very small HCOs in future national reporting and ranking.

## 5. Conclusions

The HCD database was suitable for the development of an evidence-based risk adjustment model, but showed limitations in terms of risk factors’ availability and representativeness. Risk adjustment had a substantial impact on the tertile ranking of HCOs, which underlines the need for evidence-based risk adjustment for fair comparisons of the quality of UI care in home care in Switzerland.

## Figures and Tables

**Figure 1 ijerph-18-05502-f001:**
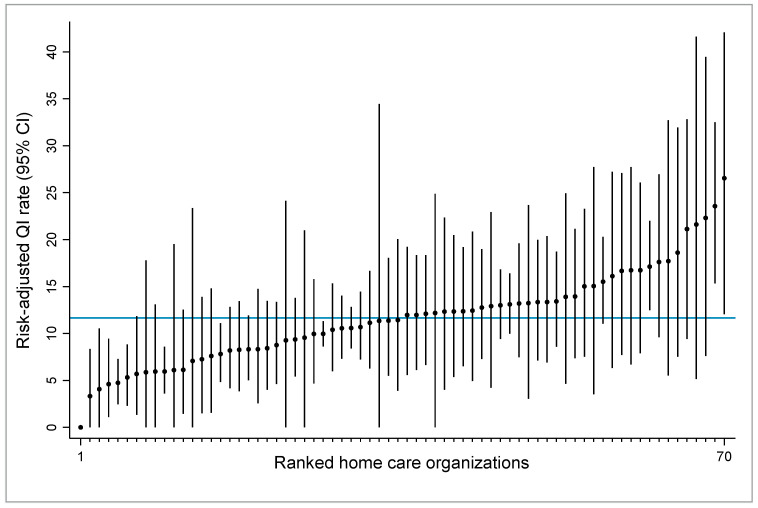
Caterpillar plot presenting bootstrap 95% confidence intervals (CIs) of risk-adjusted quality indicator (QI) rates of worsening or onset of urinary incontinence for 70 home care organizations. CIs are based on 10,000 bootstrap resamples. Organizations are ordered from 1–70 on the *x*-axis by the risk-adjusted QI rate (rank 1 corresponds to the organization with the lowest QI rate). The blue solid horizontal line shows the total QI rate (client population mean).

**Figure 2 ijerph-18-05502-f002:**
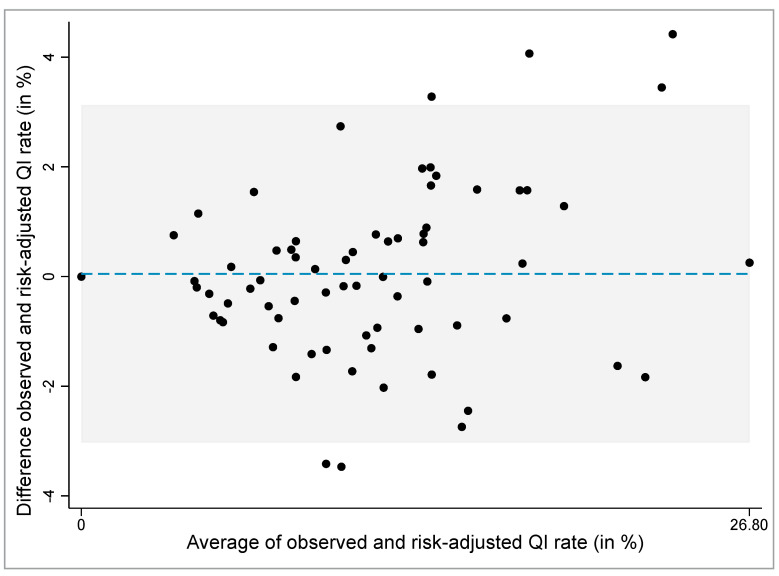
Bland-Altman plot of differences between the observed and risk-adjusted quality indicator rates. The dashed blue horizontal line shows the mean difference (0.05). The grey shaded area depicts the 95% agreement interval (−3.028 to 3.127). 8.57% (6/70 values) lies outside the limits of the agreement. The averages of observed and risk-adjusted QI rates lie between 0.0% and 26.8%.

**Table 1 ijerph-18-05502-t001:** Quality indicator bladder incontinence description.

Definition	Numerator	Denominator	Exclusion Criteria
Proportion of home care clients with worsening or onset of UI.	Number of home care clients with worsening of UI or the occurrence of new UI between two assessments.	All home care clients	Home care clients receiving palliative care and/or aged < 18 years.

Abbreviations: UI, Urinary incontinence.

**Table 2 ijerph-18-05502-t002:** Characteristics of the study population (*n* = 12,652 clients).

Characteristic		n (%)
Change in UI status ^1^	Improvement or no change	11,304 (89.4)
	Worsening or onset	1348 (10.7)
Age (years)	18–64	1557 (12.3)
	65–79	3160 (25.0)
	≥80	7924 (62.7)
Sex	Male	4571 (36.1)
	Female	8081 (63.9)
BMI	Under and normal weight (BMI < 25 kg/m^2^)	6069 (49.5)
	Overweight (BMI 25–29.9 kg/m^2^)	3631 (29.6)
	Obese (BMI ≥ 30 kg/m^2^)	2558 (20.9)
Smoking (daily)	No	11,077 (87.8)
	Yes	1542 (12.2)
CPS	0–intact	9001 (71.1)
	1–mild impairment	2560 (20.2)
	2–moderate impairment	819 (6.5)
	3–severe impairment	272 (2.2)
ADL index ^2^	Independent	11,130 (88.0)
	Not independent	1522 (12.0)
Locomotion in house	no impairment	10,847 (89.1)
	impairment	1330 (10.9)
Dressing	no impairment	8486 (67.1)
	impairment	4158 (32.9)
Personal hygiene	no impairment	8078 (64.1)
	impairment	4522 (35.9)
Bathing	no impairment	4757 (39.7)
	impairment	7221 (60.3)
Unsteady gait	No	5309 (42.0)
	Yes	7343 (58.0)
IADL scale ^3^	0–independent	1026 (8.1)
	1–supervision required	1489 (11.8)
	2–limited impairment	2168 (17.1)
	3–sometimes extensive assistance required	2418 (19.1)
	4–extensive assistance required	2285 (18.1)
	5–dependent	2487 (19.7)
	6–total dependence	779 (6.2)
DRS	0–no signs of depression	11,240 (88.8)
	1–mild signs of depression	895 (7.1)
	2–moderate signs of depression	396 (3.1)
	3–severe signs of depression	121 (2.0)
Bowel incontinence	No	11,555 (91.5)
	Yes	1077 (8.5)

Abbreviations: ADL, Activities of daily living; BMI, Body mass index; CPS, Cognitive performance scale; DRS, Depression rating scale; IADL, Instrumental activities of daily living; UI, Urinary incontinence. ^1^ Change in UI from pre-assessment to follow-up assessment. ^2^ ADL index includes: bed movement, transferring, toilet use, and eating. ^3^ IADL scale includes: meal preparation, general housework, manage money, handling medication, making a call, and shopping.

**Table 3 ijerph-18-05502-t003:** Adjusted Odds Ratios for worsening or onset of urinary incontinence (*n* = 8943 clients).

Risk Factors of Worsening or Onset of UI	OR	95% CI	
		Lower	Upper
Age (years) (ref = 18–64)			
65–79	2.247 ***	1.548	3.262
≥80	3.184 ***	2.233	4.539
Female sex (ref = male)	1.482 ***	1.269	1.732
BMI (ref = under or normal weight, BMI < 25 kg/m^2^)			
Overweight (BMI 25–29.9 kg/m^2^)	1.161 +	0.988	1.365
Obese (BMI ≥ 30 kg/m^2^)	1.396 ***	1.163	1.676
CPS (ref = 0–intact)			
1–mild impairment	1.315 **	1.111	1.557
2–moderate impairment	1.705 ***	1.321	2.201
3–severe impairment	2.279 ***	1.508	3.444
Personal hygiene impairment (ref = no impairment)	1.262 **	1.065	1.495
Bathing impairment (ref = no impairment)	1.314 **	1.099	1.571
Unsteady gait (ref = steady gait)	1.310 ***	1.120	1.532
Hospitalization in the last 90 days (ref = no hospitalization)	1.340 ***	1.144	1.571
Type of pre-assessment (ref = intake assessment)			
Regular follow-up assessment	1.152	0.963	1.378
Significant status change assessment	1.362 +	0.991	1.873

Abbreviations: BMI, Body mass index; CI, Confidence interval; CPS, Cognitive performance scale; OR, Odds ratio; UI, Urinary incontinence. Constant (intercept) = −4.28714; c-statistic = 0.658; + *p* ≤ 0.10, * *p* ≤ 0.05, ** *p* ≤ 0.01, *** *p* ≤ 0.001.

## Data Availability

The data are not available due to legal restrictions.

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
