# Peer review of "Adjusting Client-Level Risks Impacts on Home Care Organization Ranking"

_ijerph, 2021, doi:10.3390/ijerph18115502_

Round 1

Reviewer 1 Report

It seems the major problem of this work is lack of novelty, and lack of a clear method description and motivation, and also lack of a comprehensive comparison with other methods.

The paper proposed to apply client risk impact adjustment on quality indicators of the home care organizations. There are lots of weaknesses for this paper that may seriously affect the quality.

  1. I understand that it is necessary to add such an adjustment to the QI, since the unadjusted QI will not reflect the real organization home care quality because of various confounders of the clients.  However, the author needs to emphasize this point and describe the underlying reason more clearly. The authors can show some examples: it was shown that 14 of 70 HCOs shifted to another tertile after adjustment in the paper; it may be also shown why the 14 HCO rates was better after adjustment instead of before adjustment; what is the major factor that cause the 14 HCO rates to be rated inaccurately without adjustment. The authors should provide more deep insight on this problem.
  2. The risk adjustment  is not proposed by the authors for the first time. Risk adjustment for international RAI-HC QIs has been investigated. It seems the only novelty is to apply the existing adjustment method to Swiss RAI-HC QI. If that is the case, there is a lack of novelty.
  3. The paper's focus is the adjustment. However, I can hardly find the adjustment method in this paper. I can only find the author's method is "O/E ratio multiplied by the total QI rate". The author needs to give more detailed and clear description on this.
  4. The authors mentioned that there are different approaches for risk adjustment such as stratification, indirect standardization.  So why the author chose the method in the paper. Is there some specific advantage? please specify. The author would better make a comparison with those existing methods in other literatures to see if the results make any difference. The author may propose some evaluation criterion to tell us which methods are better.

Reviewer 2 Report

Thank you for giving me the possibility to review the paper “Adjusting client-level risks impacts on home care organization ranking”. This article deals with a very interesting topic, since it presents the results on the development of the risk adjustment model for the QI bladder incontinence using HCD data and the impact of risk adjustment on quality rankings of HCOs.

However, before considering it for publication on IJERPH, the following issues should be addressed:

  • Female patients represent 63.9% of the study population: may this finding affect the results of the study?
  • In the discussion session, please state how the findings of the present study could improve the home care of COVID and post-COVID patients.
  • Limit the number of self-citations to 2.

Reviewer 3 Report

The study analyzed a database concerning home care provides by health care providers retrospectively in order to identify factors that impact on qulity of care. As a surrogate for quality of care the authores choose urinary incontenence (UI).

Available follow up data allowed to judge new onset of UI.

In this huge sample eight risk factores could be identified that had an impact on the onset of UI.

Adjustment for these factors lead to a change in quality ranking of the single health care providers.

The main result of this analysis is that such factors must be considered before any ranking or benchmarking concering health care providers is done.

Methodes and limitations of the analysis are discussed comprehensively.

But: there seems to be a problem conrening reference sources. This problem should be solved easily

Round 2

Reviewer 1 Report

The authors' modifications now have summarized more clearly their contributions.  The author's work is more useful in evaluating home care with many small providers (like Switzerland) with adjusted quality rankings, which have not been studied by existing literatures. Specifically I have the following comments:

  1. On page 5, line 226, Page 3 Line 140, Page 7 Line 246, there are some typos on the references "Error! Reference source not found." please correct.
  2. There are lots of typos. Please carefully check all typos throughout the paper. For example,  in  Section 4.2 Line 386, "Altough", Line 388 "followig" Line 389 "soluation", Line 245 "multivariable logistic regression". line 250-251, "The c-statistic in the validation data set was not significantly different with a value 250 of 0.685 " should this be 0.685 or 0.658 (c-statistics of the training data)?
  3. for the logistic regression in the section 2.4, the author selected covariates with a p value ≤ 0.25 in univariate analyses first, and then in the joint model risk factors that were non-significant (p > 0.1) were excluded.  Here are two tuning parameters 0.25 and 0.1. If you change those parameters how will influence the model performance such as the c-statistics on validation set, and how will it influence the final rankings? It's better to check the robustness against tuning parameters. Or give a specific reason to choose such parameters.
  4. Is there some specific references for the selected risk factors here? For example, I can see that the most important predictor is age. Age now was included as a categorical predictor in the model by using "[65,79]" ">=80". Is there some specific reasons to use 65 and 80 as cutoffs or will it make any difference if you use the continuous age variable?
